# Transients as the Basis for Information Flow in Complex Adaptive Systems

**DOI:** 10.3390/e21010094

**Published:** 2019-01-20

**Authors:** William Sulis

**Affiliations:** Department of Psychiatry and Behavioural Neuroscience, McMaster University, Hamilton, ON L8N 3K7, Canada; sulisw@mcmaster.ca

**Keywords:** information, semantics, salience, complex adaptive systems, transients, TIGoRS, Sulis machines

## Abstract

Information is the fundamental currency of naturally occurring complex adaptive systems, whether they are individual organisms or collective social insect colonies. Information appears to be more important than energy in determining the behavior of these systems. However, it is not the quantity of information but rather its salience or meaning which is significant. Salience is not, in general, associated with instantaneous events but rather with spatio-temporal transients of events. This requires a shift in theoretical focus from instantaneous states towards spatio-temporal transients as the proper object for studying information flow in naturally occurring complex adaptive systems. A primitive form of salience appears in simple complex systems models in the form of transient induced global response synchronization (TIGoRS). Sparse random samplings of spatio-temporal transients may induce stable collective responses from the system, establishing a stimulus–response relationship between the system and its environment, with the system parsing its environment into salient and non-salient stimuli. In the presence of TIGoRS, an embedded complex dynamical system becomes a primitive automaton, modeled as a Sulis machine.

## 1. Introduction

Science is slow to accept new concepts and ideas, but once established, they gain widespread usage. Matter, wave, and energy are such concepts. Information may be the latest judging by a recent spate of books [1,2,3]. The word information itself dates back to the 14th century, but related concepts such as meaning go back millennia. The modern formalized conception of information takes four main forms, both arising in the early to middle years of the 20th century.

The first, Fisher information, comes from probability theory. Suppose that the probability distribution for a random variable *X* is modulated by some additional parameter *θ*. One would like to know how well a measurement of *X* predicts the value of *θ*. The *score* is defined as the derivative with respect to *θ* of the natural logarithm of the probability distribution,
$$Sc=\frac{\partial }{\partial \theta }\log f(X;\theta )$$

*Fisher information* is defined as the variance of the score
$$I(\theta )=\int (\frac{\partial }{\partial \theta }\log f(X;\theta ){)}^{2}f(X;\theta )dX$$
and it is mostly used in statistical theory.

The second approach to information was developed by Shannon in the context of communication. Shannon considered communication channels carrying sequences of symbols. These symbols could be letters, dots and dashes, images. In the simplest case, one assumes that they are finite in number and can be denoted {*N_i_*} and that they are transmitted by an ergodic Markov process, so that the statistical properties of almost all generated sequences are identical. If *p_i_* is the probability of occurrence of the *i*-th symbol, then the *average amount of information* carried by the channel is defined as
$$H=-\Sigma_{i}p_{i}\log p_{i}$$

Shannon’s notion of information bears a striking similarity to the physical concept of *entropy*, and indeed von Neumann suggested that Shannon use the term entropy when referring to this in his paper. Prior to Shannon, the concept of information played a limited role in physical theory. In the decades following Shannon’s definition of information, the concept of information has come to play an increasingly prominent role in physics, particularly in quantum foundations. The introduction of information theoretic ideas into quantum mechanics spawned the creation of the sub-discipline of quantum information, and that in turn generated an interest in ideas of quantum computing. Information has gradually been accorded the status of a physical construct with physical properties. Shannon was explicit in pointing out that his theory was only concerned with the capacity of engineered systems to reliably convey signals from a sender to a receiver [4] (pp. 31,32): “The fundamental problem of communication is that of reproducing at one point either exactly or approximately a message selected at another point. Frequently the messages have *meaning*; that is, they refer to or are correlated with certain physical or conceptual entities. These semantic aspects of communication are irrelevant to the engineering problem. The significant aspect is that the actual message is one *selected from a set* of possible messages. The system must be designed to operate for each possible selection, not just the one which will actually be chosen since this is unknown at the time of design.

If the number of messages in the set is finite then this number or any monotonic function of this number can be regarded as a measure of the information produced when one message is chosen from the set, all choices being equally likely (SIC)”.

The content of said information, its meaning, was never a consideration for Shannon, who was primarily concerned with what were essentially the universal aspects of communication systems. Over time, the entropic characterization of information and its variants has become the definition of information. Fisher information also lacks any reference to meaning.

The third approach comes from the formal theory of computation. Here, information is represented as data, usually in the form of discrete strings of formal symbols, which is fed into some formal computational device, which then in turn generates a second symbol string corresponding to the outcome of the computation. Meaning per se is again ignored, although such data does make a difference in the way in which its presence alters the activity of the computational device. The emphasis is placed upon finding and understanding the universal rules of computation, by which any such information can be usefully transformed. The theory of computation has focused mostly on mathematical computations and formal models of digital computers. Biological computation has mostly been explored in terms of neural networks. One of the central notions of the theory of computation is the distinction between the data and program, and the dual nature of symbol strings. Another is the distinction between recursive, recursively enumerable, and non-enumerable symbol strings, or more simply, computable versus non-computable.

The fourth approach also lies within the realm of computation. It is called algorithmic information theory and measures the information (or complexity) of an object as the size of its smallest description.

These formal theories have focused upon concepts that are universal, optimal, parsimonious, rational, logical, ideal. They generally assume perfect knowledge, perfect information, infinite resources, and where resources are limited, their optimal exploitation. To distinguish between these formal idealizations and the realities of complex adaptive systems as they manifest in nature, the concept of the naturally occurring computational system (NOCS) was developed [5].

The idea of a NOCS is to capture the formal dynamical features of behaving systems as they occur naturally in the real world, as opposed to those systems that have been artificially or theoretically created by human beings. A NOCS exists in the real world and its main goal is to survive, generally by whatever means suffice. It lives in a constantly changing environment with which it continually interacts. It responds to its environment and in turn it acts upon its environment, modifying it in myriad ways. It possesses agency, interprets events, forms intentions, and acts. It suffices that its actions be “good enough”. It never possesses complete information, seldom has the time to examine all possible choices before making a decision, and it uses resources without necessarily concerning itself with their extent.

As an example, consider a migratory bird. During the course of a year it is faced with at least three central tasks: mating, nesting, brood rearing; migration; and maintenance foraging. Mating requires finding a suitable mate, and the strategies for doing so will depend upon the phenotype of the bird, its habitat, food resources, and the availability of mates. In environments with abundant conspecifics, it can afford to be choosy, and may seek a mate based upon some optimality criteria. When mates are scarce it may adopt a satisficing strategy, and accept the first mate that it encounters. During nesting and brood rearing many tasks must be carried out such as foraging, feeding, and defense. The environments will all differ from season to season. No bird is born with knowledge of all the possible locales, and all the possible environmental conditions. Knowledge is acquired “on the fly”, and decisions must often be made in a split second, without time to explore all the possible options. In defending itself from attack, a bird will not necessarily seek the optimal response; it will generally settle for a response which is “good enough” and allows it to escape and survive. Migrations may be conducted individually, in flocks or in swarms, with each exhibiting quite distinct dynamics. Migrations often extend over many weeks and hundreds or thousands of kilometers, across widely varying terrain.

A bird exists in an open relationship with its environment. It not only consumes and transforms energy, it also transforms itself. Its basic constituents are constantly in flux. Its physical form is constantly changing through development and aging. Not only does it instantiate through its behavior some dynamics within a phase space, but the phase space itself is dynamic. Through its behavior, a bird expresses functionality. Its behavior does something, it makes a difference: foraging, eating, feeding, nest building, rearing, flying, and so on. Behaviors serve a purpose in the life of a bird. Moreover, behaviors are not defined by instantaneous states but rather by patterns that extend over some duration (and sometimes space). Behaviors are spatio-temporal transients.

NOCS are emergent, arising out of interactions among a collective of lesser entities. In general, patterns of behavior are not programmed, they arise spontaneously from these interactions and are modified by learning. Moreover, behaviors which appear similar at the macroscopic level are, in general, generated in different ways at the mesoscopic or microscopic levels. For the most part, every behavioral act (including memories) is generated anew. Such is the case of multi-cellular organisms. So too is the prototypical example of a NOCS, widely studied observationally, experimentally, and theoretically, a collective intelligence system, in particular, social insect colonies. Their dynamics possess several important features [6,7,8] which are not usually considered in the models of computation:
Self-Organization.Stochastic determinism.Interactive determinism.Nonrepresentational contextual dependence: knowledge within a collective intelligence is nonrepresentational. The environment carries the information that a collective intelligence requires.Phase transitions, critical, and control parameters.Broken ergodicity.Broken symmetry.Pattern isolation and reconfiguration refers to the appearance of dynamical regularities amongst the dynamical transients exhibited by a system, such that these transients can be identified as entities or as states of entities in their own right, having their own temporal evolution, and patterns of action and interaction.Salience refers to the identification of dynamical stimuli that are capable of influencing the dynamical behavior of a system in a meaningful and consistent manner. Salient stimuli produce dynamically robust and stable effects. Conversely;Irrelevance refers to the situation in which specific features of the dynamical behavior of a system at one level, say for example the functional form of a phase transition curve, do not depend in any meaningful manner upon knowledge of the dynamical behavior at lower levels, such as the specific nature of microscopic interactions. Saliency and irrelevance play opposite roles in understanding the relationships between the dynamics of individuals and the dynamics of the collective.Compatibility and the mutual agreement principle [9] refers to the notion that interactions between the individuals of a collective are not always random, but they frequently involve a choice to interact or not which depends upon extrinsic factors that are salient to the individuals, and an interaction does not occur unless both parties agree.

NOCS may engage in direct physical interactions with one another and with their environment. They may also interact with intermediaries, lesser physical entities such as patterns of light, acoustic waves, odorants, etc., which serve as signals. Their purpose is to enable the sender to influence the behavior of the receiver in some ecologically significant manner. The sender usually sets the form of the pattern, so as to convey a particular meaning. The receiver must then interpret this pattern and, hopefully, obtain the same meaning as the sender intended. Both the sender and receiver construct and interpret these patterns relative to their own semantic frames [10]. These frames are generalizations of the physical frames of reference applied to the realm of semantics, and their study is described as archetypal dynamics [10]. Semantic frames establish which patterns are salient to the NOCS, and they shape the reaction that should occur in response.

Semantics, the study of meaning, is a vast field with a long history. There have been efforts to develop a theory of semantic information [11] as an addition to the Shannon information theory. The field of semantics more generally has developed its own approaches. In particular, the exploration of the role of meaning among biological entities has spawned the field of biosemantics.

Meaning plays a fundamental role in mediating interactions between a NOCS and its environment. Meaning appears in the interaction between a stimulus and a recipient. The responses of a NOCS to a signal can be remarkably adapted to their ecological niches. Among ants, certain pheromones convey a universal meaning as a signaling alarm. The response to this pheromone is not universal in its form. For example, the release of an alarm pheromone by an individual of a colony of *Acanthomyops claviger*, a subterranean dweller, results in the rapid assembly of many workers, all agitated, mandibles ready, at the site of the disturbance. In *Lasius alienus*, whose colonies normally reside under rocks or in rotting wood, the release of the alarm pheromone causes nearby workers to scatter widely. The stimulus is the same for both species, yet the response appears to be tailored to the specific needs of each species [12] (pp. 236,237).

## 2. The Role of Transients in NOCS

What is the natural level of description for information in complex adaptive systems? This is not clearly specified in either the Shannon or computational models of information. In the Shannon model, one has some channel along which is being transmitted a sequence of discrete signifiers, such as a,a,b,c,d,e,b,b,…. In a computational model, there are two signifier sequences, an input sequence a,a,b,c,d,e,b,b,…. and an output sequence r,s,w,x,t,t,t,t,… No time signature is attached to any of the signifiers in these sequences. They may represent instantaneous events. They may equally well represent events having a temporal extension, a duration.

Collective intelligence systems, the quintessential examples of a NOCS, utilize a variety of signals to pass information among their members. These signals are not instantaneous, rather they extend over several seconds to minutes and they often possess a complex spatio-temporal structure. The most famous of such signals may well be the waggle dance used by Honeybees to send foraging information to their nest mates. The waggle dance does not consist merely of isolated moves, but instead has a rich structure with relevant information encoded in particular sequences of moves [12].

Another example which has been studied in exacting detail is the process of nest selection and emigration in colonies of the ant species, *Temnothorax albipennis*. This minute ant species (workers are about 1 mm in length) live in small colonies (100–500 workers) in nests established in small crevices in cliffs. These nests are subject to frequent collapses, and so the search for new sites and subsequent emigrations are frequent occurrences. Colonies are capable of evaluating a number of factors, such asthe degree of light, heat, humidity, exposure to threat, spatial expanse, elevation, and so on, when selecting potential nest sites. When the nest is disturbed, individual workers (scouts) are stimulated to explore the surrounding environment for suitable sites for a new nest. Scouts explore the local environment initially at random. When a scout encounters a potential site, it assesses the site according to several criteria which are integrated into a quality measure. This quality measure is expressed as a wait time before recruitment that varies inversely with quality. Once induced to recruit, scouts return to the original nest and encourage fellow nest mates to attend their choice of a new nest site through tandem running. In tandem running, the nest mate is induced to run alongside of the scout and thereby follow them back to the chosen site. This process of recruitment results in a gradual increase in the number of workers within the new site. When a threshold in numbers is achieved or exceeded (quorum threshold), the scouts will shift strategy and initiate social carrying, which brings large numbers of nest mates to the new site [13].

Individual workers do not choose by comparing alternative sites [14]. Instead, their choice appears to be via a threshold rule which alters the length of time during which they remain at an alternative site before returning to the original nest [15]. That is, when they encounter a preferred site they tend to remain at the site for a prolonged period of time and when they do leave they tend not to visit other sites. On the other hand, if they encounter a non-preferred site they linger for a much shorter period of time and they tend to explore other available sites. Thus, they exhibit distinctly different forms of transient behavior depending upon the quality of the site that they encounter. Interestingly, individual workers appear to have different thresholds [16] and different durations of wait times. This heterogenicity appears to confer an advantage when the choices of alternative nest sites are below the acceptance thresholds, and in environments in which attributes are fluctuating. In particular, the transient behavior associated with lingering has a significant impact on the effectiveness of the decision making at the colony level.

Released pheromones also tend to linger in a region over seconds to minutes depending upon the environmental conditions, and it is the persistence of these pheromones that gives rise to their efficacy in stimulating subsequent behavior [12]. The diffusion of pheromones is also a dynamical transient. Stimulation of nest mates via direct contact also extends over a period of time, and thus, it is again a dynamical transient.

Another form of signaling is through stigmergy, the incitement to work by the products of work [12]. Many social insects act upon their physical environment, transforming it to meet some ecologically salient purpose such as nest construction. Many ant and wasp species will construct complex nests. No individual worker contains the blueprint for such a nest. Instead, the workers actions are triggered by the physical objects created by the prior actions of other workers. This on-going interaction between the workers and a changing environment results, over time, in the construction of a species-specific nest structure. This is another example of an emergent process in which spatio-temporal transients (here the individual structural fragments) serve as the stimuli for subsequent behavior.

This very brief survey demonstrates that many salient forms of signaling in the NOCS appear to involve spatio-temporal transients. Any realistic model of a NOCS must therefore consider the role of these spatio-temporal transients.

Moreover, these colonies show a surprising subtlety and complexity in their decision making. Reference [13] showed that when presented with several alternatives, the colonies effectively executed an additive weighted strategy, a rational strategy which is difficult even for humans to utilize. Moreover, Edwards and Pratt [17] and Sazaki and Pratt [18] showed that decision making at the colony level could be rational, even when decision making at the level of the individual worker was non-rational. They showed that individual workers were subject to the decoy effect, a violation of the principle of regularity which states that a preference should not be altered by the appearance of a non-preferred option. They also showed that the colony to which these same workers belonged did not exhibit the decoy effect when presented with the same options and decoys.

In addition, these colonies appear to be capable of adapting their decision-making strategy under differing environmental conditions [19]. Thus, these colonies are capable of rational and bounded rational decision making under different conditions and they appear to be capable of learning [20].

Colonies of *Temnothoraxalbipennis* implement their decision making in an emergent manner through the collective actions and interactions of their individual workers. This is strikingly different from the manner in which individual humans make decisions. Thus, social insect colonies and humans can implement similar decision making albeit in a markedly different way. Several years ago [21,22], the dual notions of computational competence and computational performance were proposed. Computational competence refers to whether or not a system has the ability to perform or support a specific computation. Computational competence addresses the question ‘what’? Computational performance refers to the exact process through which this specific computation is implemented.

Computational performance addresses the question of ‘how’? Computational competence is the primary concept and computational performance is the secondary. Much of the theory of computation has focused upon questions of computational performance. However, in dealing with a NOCS, the more fundamental question is that of computational competence. Exploring that question leads to approaches to modeling that are based upon the ideas of function and process. These ideas require novel mathematics, since they involve notions of transience, fungibility, generativity, development, and flux, all of which create profound challenges for the orthodox approaches. These ideas have been developed in the theories of functional constructivism and di-evolution [23], and in process algebra [24].

Another theory of more direct relevance here is that of the dynamical automaton [21,22], later described as Sulis machines [25].

Assume that we are given a dynamical system (Y × Ω, *T*, *ϕ*), and that the monoid *T* is totally ordered. We form the totally ordered monoid *I_T_* consisting of all intervals of the form [0,*q*), where 0≤q∈T. The semigroup operation is given by concatenation, [0,*q*)[0,*p*)=[0,*p*+*q*),and the identity is the empty interval [0,0). We form two new monoids, Y(*I_T_*) and Ω(*I_T_*), consisting of all maps from the elements of *I_T_* to Yand Ω, respectively. That is, an element of Y(*I_T_*) consists of a map from some interval [0,*t*) in *I_T_* toY. The monoid operation is given by concatenation. For example, given maps $[0,p)\overset{\psi }{\mapsto }\Upsilon$ and $[0,q)\overset{\rho }{\mapsto }\Upsilon$, we can define a map *ψρ* from [0,*p*+*q*) to Y by ψρ(y)=ψ(y) if y∈[0,p), and ρ(y−p) if y∈[p,p+q).

The empty map 0 defined on the empty set [0,0) yields the identity. For any element *ψ* of Y(*I_T_*) (likewise Ω(*I_T_*)), we may assign a value *μ*(*ψ*) in *T* given by μ(ψ)=t if $[0,t)\overset{\psi }{\mapsto }\Upsilon$.

We define a dynamical automaton as (Y(*I_T_*), Ω(*I_T_*), Δ), where the transition function Δ from Y(*I_T_*) × Ω(*I_T_*) to Y(*I_T_*) satisfies the following:
(1)Δ(0,ρ)=0(2)$\Delta (\psi ,\rho )=\psi \psi^{\prime }$(3)$\mu \psi^{\prime }=\mu \rho$(4)Δ(ψ,0)=ψ(5)$\Delta (\psi ,\rho \rho^{\prime })=\Delta (\Delta (\psi ,\rho ),\rho^{\prime })$

The first two conditions arise because we are dealing with the transient orbits of systems rather than individual states, and so they are necessary to maintain consistency and timing.

Definition of Computation in Dynamical Automata:

Let A = (Y(*I_T_*), Ω(*I_T_*), Δ), be a dynamical automata. An instance of an information process is a pair *P* = (*W_Q_*, *W_R_*), where *W_Q_* is an open, bounded subset of termed the question, and *W_R_* is an open, bounded subset of Y(*I_T_*), termed the response. The automaton *A* processes *P weakly* if there exists states ψ ∈ Y(*I_T_*), *ρ* ∈ *W_Q_*, and *ϕ* ∈ *W_R_*, such that Δ(ψ,ρ)=ϕ. The automaton *A* processes *P effectively* if there exists an open, bounded subset $W_{E}\subset \Upsilon (I_{T})$, and a subset ${W^{\prime }}_{Q}\subset W_{Q}$, such that $\Delta (W_{E},{W^{\prime }}_{Q})\subset W_{R}$. The automaton *A* processes *P strongly* if there exists a subset ${W^{\prime }}_{Q}\subset W_{Q}$, such that $\Delta (\Upsilon (I_{T}),{W^{\prime }}_{Q})\subset W_{R}$. The automaton *A* processes *P error free* if ${W^{\prime }}_{Q}=W_{Q}$.

The open sets of the dynamical automaton provide a formal analogue of the concept of behavior in psychology. These sets consist of correlated spatio-temporal transients, just as behaviors consist of spatio-temporal transients generated by an entity, linked by some criteria which may be structural or functional.

Transients appear to play a role in information processing in the collectives of agents, such as social insect colonies. They play a role in the dynamics of individuals as well. Freeman [26], in his pioneering study of mesoscopic brain dynamics, emphasized the role of transience, and therefore of transients, in information processing in the brain. He was particularly interested in the role that chaotic dynamics and strange attractors might play in such information processing. His seminal work showed the existence and importance of chaotic dynamics within the brain. Based upon extensive studies of sensory processing in insects, he conjectured that information processing might involve a succession of transitions between strange attractors, each of which correspond to patterns of behaviors that are associated with particular meanings. The time spent within each strange attractor, being finite, resulted in the generation of a spatio-temporal transient. Information processing is observed in the succession of these spatio-temporal transients, determined by the sequence of shifts between the strange attractors. Interactions with the environment trigger these shifts among the strange attractors, and the resulting spatio-temporal transients (behaviors) exhibit consistency in either the structure or functionality reflective of meaning. All of this complexity arises in an emergent manner from the underlying neurodynamics. The dynamical automaton provides a simplified model of these dynamics.

## 3. Information in Formal Complex Adaptive Systems

Modeling of complex adaptive systems within mathematics, physics, and engineering has focused attention on either the instantaneous states or the asymptotics of trajectories. For the most part, this has been true in biological modeling as well, even though as described above, the transients play an important role. The phenomenon of synchronization has been given some prominence in biological modeling, but this research has again focused on the phase locked synchronization of the trajectories of distinct entities, be they individual cells or entire organisms [27]. The study of the behavior of transients in complex adaptive systems began in a series of papers and presentations [21,22,28,29]. These focused on a particular phenomenon involving transients termed transient induced global response synchronization (TIGoRS).

Most models of complex systems have treated the systems as being closed and isolated. NOCS, on the other hand, are open systems that are deeply embedded within their natural environment. Therefore, formal models of a NOCS must be open and embedded as well. The above papers were among the first to study complex systems models in an open setting, interacting with an environment.

Several complex systems models (tempered neural networks, driven cellular automata, coupled map lattices, cocktail party automata, dispositional cellular automata) were studied under the conditions of external perturbations, or stimuli, from an environment. Application of transient stimuli to individual cells did not, in general, result in a phase locked synchronization of their trajectories. However, it was quickly discovered that when a transient stimulus was applied to the individual elements of a complex system, regardless of the initial state of the state of the system, the resulting global transient state patterns clustered together in the pattern space. This was the first demonstration of TIGoRS. In the original work, TIGoRS was said to be present if two trajectories under a stimulus were significantly closer in Hamming distance than would be possible by random chance.

In each of these cases, a class of complex systems having identical numbers of agents was studied. A stimulus pattern was first created by letting a single complex system evolve and then selecting a transient trajectory, usually of 450 time steps. Low frequency, random samples of this pattern were then created. A second complex system from this class was simulated from different initial conditions with different samples applied as the external stimuli. A given stimulus was applied to corresponding agents at corresponding times and usually involved setting the state of the stimulated agent to match that of the sample. Sample frequencies ranged from 1% to 100%. It was noted that in a fairly wide range of systems, sampling frequencies as low as 5% could produce response transients that matched by as much as 95%.

In many cases, this matching occurred after a brief initial transient, but in other cases, there were always a few agents which differed from one another no matter how long the duration of the stimulus. In all cases, the matching was between the global response patterns, that is, with entire state space transients having the same duration as the stimulus. Individual agents within the system did not synchronize. The synchronization was global between the stimulus transient and the response transient (see Figure 1). TIGoRS appeared to be stable in the sense that a stimulus sample pattern could be presented followed by a random stimulus, then followed again by the stimulus pattern and the same response occurred. Importantly, TIGoRS was not universal across all stimulus patterns. Instead, only select patterns appeared to be able to induce TIGoRS and these patterns varied among the different complex systems. Thus, each class of complex system appeared to partition the space of the stimulus patterns into those that were salient and those that were not. This assignment of salience served as a precursor to the establishment of meaning.

In the case of cellular automata, the degree of concordance between the stimulus and response across different initial conditions depended upon the sampling rate, the spatial distribution of the sample process, and the symmetry class of both the cellular automaton and the pattern. The symmetry class reflected the dominant symmetry that was present in the autonomous patterns produced by the automaton. The classes were uniform, linear, diagonal, complex, and chaotic [28] in order of decreasing symmetry.

A more refined measure of TIGoRS was introduced in Reference [27]. The efficacy of TIGoRS was defined as E(p)=−P+H(0)−H(p), where *P* is the percentage of pattern cells sampled, *H*(0) is the percentage Hamming distance between the current automaton response and the template response in the absence of input, and *H*(*p*) is the percentage Hamming distance between the template response and the current automaton response under input by pattern *p* with sampling *P*. Efficacy and the Hamming distance curves, with and without TIGoRS, are presented in Figure 2 and Figure 3.

In the absence of TIGoRS, the efficacy curve is strictly decreasing and negatively valued. The decline in the Hamming distance with increasing sampling frequency is entirely due to the stimulus alone, which induces matching through the replacement of state values with pattern values. In the presence of TIGoRS the curve is nonlinear, with an initial positive segment reflecting the effect of the internal dynamics of the system on pattern production, followed by a more linear, decreasing segment during which the effect of the stimulus itself comes to dominate the dynamics. In the presence of TIGoRS, the system dynamics itself acts to complete the stimulus pattern; hence, the very rapid decline in the Hamming distance over low stimulus sampling frequencies.

Several models (tempered neural networks, driven cellular automata) all possessed agents following fixed rules. In Reference [29], an adaptive cellular automaton, the cocktail party automaton, was introduced. The cocktail party automaton is a finite, deterministic, uniform, but inhomogeneous, cellular automaton in which the rules vary not only across the cells of the automaton but also with time. The rule for a given cell changes according to a nonlocal algorithm, which considers the response of a majority of cells having the same neighborhood configuration. Briefly, at time *t*, cell *A* with neighborhood *N* updates its state to *s* and then checks throughout the lattice to observe which states were chosen by those cells where the neighborhoods had the same values as *N*. If the majority of cells chose state*s’*, then the cell updates its function to assign *s’* to the neighborhood value *N*. The TIGoRS phenomenon was even more pronounced in such adaptive automata. This was true even though the final rule configuration of the cocktail party automaton varied with the initial state and rule configuration and with each sampling of the stimulus.

In Reference [27], a new class of adaptive automaton, the dispositional cellular automaton, was introduced to eliminate the non-locality of the cocktail party automaton, while at the same time capturing some of the essence of G-protein coupled receptors, which are dominant in brain dynamics. The effect of these receptors is to alter the dynamics of the stimulated neurons without directly inducing the initiation of action potentials.

The dispositional cellular automaton has the same basic structure as a standard deterministic cellular automaton with the addition of a rule updating rule. If at time t a cell a with a previous neighborhood A receives an external stimulus *x*, then its state is reset to the value *x*, and its rule is modified so that it now produces the state *x* whenever its neighborhood is *A*.

Figure 4 shows the Hamming distance curves for a driven cellular automaton, cocktail party automaton, and a dispositional cellular automaton under the same stimulus pattern.

The presence of TIGoRs establishes a stimulus–response pairing of the kind required for computation in a dynamical automaton. This demonstrates the possibility of information processing at the level of global transients.

## 4. Discussion

At first glance, the idea that transients play a fundamental role in information processing seems almost obvious, at least to anyone with even a passing awareness of biology and psychology. Every behavioral act constitutes a spatio-temporal transient. The life of an organism is expressed in a never-ending succession of finite duration, spatially limited, behavioral acts. These behavioral acts show consistency in relation to particular environmental events, which are thought to elicit them. They show much less consistency when compared one to another without reference to their environment.

It is remarkable then, that virtually no attention is paid to the role of transients in the formal study of information and computation, and of dynamical systems more generally. Dynamical systems are generally studied in isolation, as closed systems. The environment appears either as a small random perturbation or as a periodic, predictable stimulus. Long term, asymptotic behavior is the focus of study. Transient behavior is dismissed as the uninteresting behavior which precedes the appearance of the interesting asymptotic behavior. The components, if any, of a dynamical system are considered to be fixed, as is the phase space within which the dynamics play out. Identical initial conditions result in identical behavior. Nearly identical initial conditions result in nearly identical behavior (stability), or when instability occurs, it generally has a structured form (chaos).

The discussion of a NOCS shows how removed from the reality of natural behaving systems these dynamical systems models are. NOCS are complex multi-agent adaptive systems. Their dynamics is emergent from interactions among their constituents and their environment. They are open systems in the extreme in that matter, energy, and information are in constant flux. The NOCS fundamental constituents change in composition, form, and network structure over time. Behaviors that appear identical at the emergent level may be generated in entirely different ways at the constituent level. Initial conditions that appear identical at the emergent level may result in very different emergent level behaviors, or in behaviors that appear similar in the large, but differ in the small, or which subserve the same functionality. Adaptation may result in repeated stimuli eliciting different responses depending upon what has happened in between their presentations. On the other hand, some responses may be remarkably robust and recur regardless of what intervenes. Thus, the stability of the responses is dependent upon the stimulus. Asymptotic behavior is irrelevant, since the environment is constantly changing. Moreover, NOCS possess agency and they act to change the environment themselves, often to satisfy particular needs and facilitate particular functions.

Novel concepts, such as dynamical automata, functional constructivism, di-evolution, and process algebra are directed at correcting the serious shortcomings in the standard dynamical systems models. Dynamical automata and Sulis machines represent a first step in directly addressing the role of transients in information flow in the complex systems. The discovery of TIGoRS, and its realization in a wide range of dynamical models (tempered neural networks, driven cellular automata, cocktail party automata, dispositional cellular automata, coupled map lattices) shows the ubiquity of the phenomenon. In TIGoRS, there need be no synchronization of activity between the individual constituents on a state by state basis. Instead, the synchronization is between a transient representing an environmental stimulus and the global response from the system. It is a correspondence between spatio-temporal transients, not between the individual states. Moreover, TIGoRS can occur even though the actual stimulus is a random, low frequency sampling of the environmental transient, such that the individual constituents receiving the stimulus vary from trial to trial. Even more, TIGoRS can occur even though the constituents possess different dynamics themselves, and these dynamics change as a result of experience (as in the case of the cocktail party and dispositional cellular automata). In the transient scenario, we see exactly the kinds of openness (or contextuality), emergence, generativity, and fungibility as seen in the NOCS.

The simple experiments of TIGoRS in the dispositional cellular automata showed that TIGoRS is a genuine nonlinear effect in these complex systems, and not merely an epiphenomenon of the stimulation process. Whether the TIGoRS phenomenon can serve as a viable model for information flow in a NOCS requires further investigation, both theoretical, as well as empirical. There is a need to explain theoretically how it is that TIGoRS arises in these open systems—what determines it, how to characterize it, how to quantify it. There is a need for more simulations, but this is hampered by the sheer combinatorial complexity associated with transients. The pattern space for even 450 time step transients of the kind used in this study was vast, (2^100^)^450^ possible patterns. There were 256^100^ possible rule configurations per time step, and so (256^100^)^450^ rule trajectories. Simulations clearly cannot explore the totality of such a space, but theory guided explorations can shed considerable light on the mechanisms that are involved in TIGoRS. Meanwhile, the naturalistic observation of NOCS, especially collective intelligence and neural systems, might demonstrate whether or not TIGoRS like phenomena occur in these systems, as well as their prevalence.

An interesting question posed by an anonymous referee was whether or not there was any role for entropy in TIGoRS. In several papers [27,28,29], a variety of different stimulus patterns were sampled during these trials. Different complex systems models appeared to respond to different symmetry types. In some cases, patterns with very high levels of symmetry induced TIGoRS. These were low entropy patterns. In other cases, complex, chaotic, or even pseudorandom patterns induced TIGoRS. These were all high entropy patterns. It was not clear whether there is any relationship between the entropy of a pattern and its ability to induce TIGoRs, or the entropy of a pattern generated by an autonomous complex system and that of the patterns that induce TIGoRS in it. This would seem to be an interesting area for future study.

## Figures and Tables

**Figure 1 entropy-21-00094-f001:**
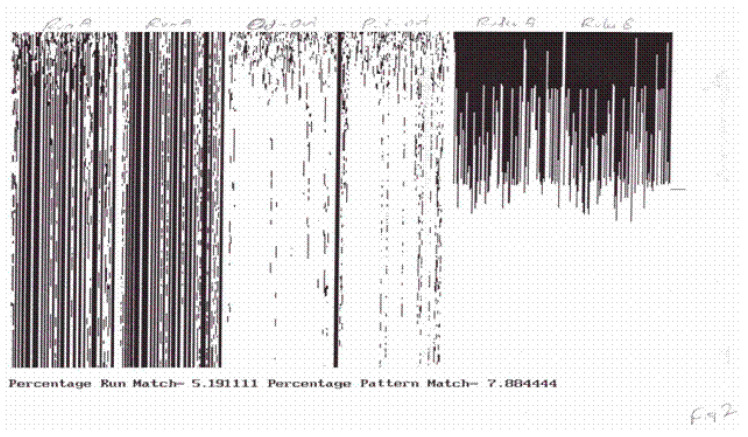
The pictures illustrate transient induced global response synchronization (TIGoRS) in a cocktail party automaton. The first two pictures show individual runs under different initial conditions and different low frequency samples of the same stimulus pattern. The third picture shows discordance between the first two runs. The fourth picture shows discordance between the first run and the stimulus. The final two pictures show the distributions of the rules at the end of each run.

**Figure 2 entropy-21-00094-f002:**
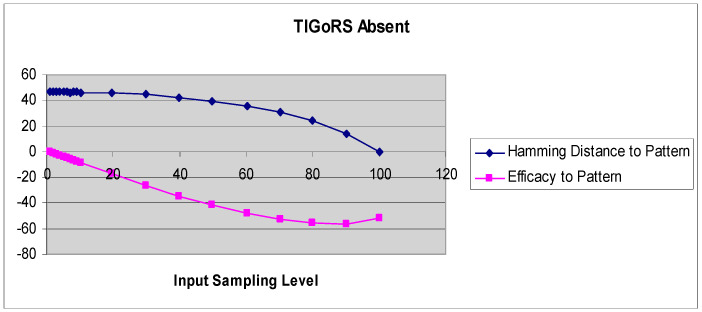
The graphs depict the Hamming distance between the stimulus and response and efficacy as a function of the stimulus sampling rate in the absence of TIGoRS.

**Figure 3 entropy-21-00094-f003:**
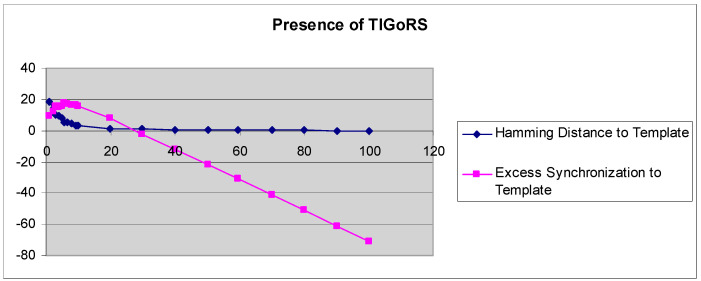
The graphs depict the Hamming distance between the stimulus and response and efficacy as a function of the stimulus sampling rate in the presence of TIGoRS.

**Figure 4 entropy-21-00094-f004:**
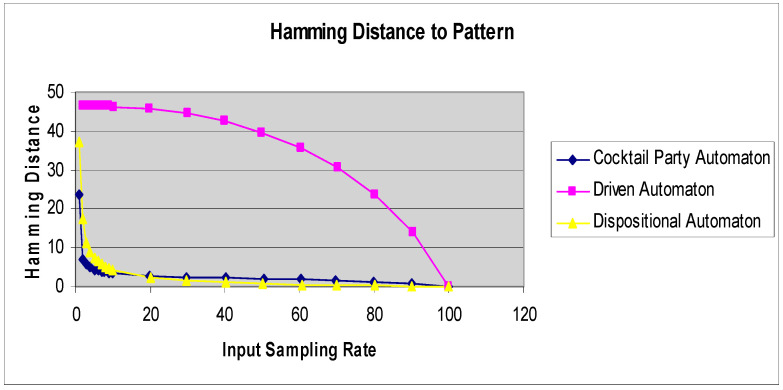
Hamming distance curves for three different automaton classes under the same stimulus.

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
