# Peer review of "Transients as the Basis for Information Flow in Complex Adaptive Systems"

_entropy, 2019, doi:10.3390/e21010094_

Round 1
Reviewer 1 Report
Very well done. Would it be of any value to identify some information transients in mammalian or human signaling that leads to "stable collective responses"?
Author Response
Thank you for your review. In response to your concern, an additional paragraph has been added to the end of section 2, The Role of Transients in NOCS, which describes briefly the work of the famed neurobiologist Walter Freeman. Freeman was a pioneer in the study of nonlinear dynamics and chaos in nervous systems, both human and insect, especially locusts. His research into mesoscopic dynamics emphasized the role that chaotic dynamics and strange attractors play in psychological processes, including meaning. He wrote of shifts between these attractors, which has the effect of partitioning trajectories into dynamical transients.
I hope that this example suffices, but I can provide more if required.
Reviewer 2 Report
The paper ”Transients as the basis for information flow in complex adaptive systems” proposed by author is interesting and follows briefly a research trajectory and direction concerning the flow of information in complex adaptive systems, focusing on the role, importance and model of transients.
The author proves good knowledge of the proposed research, determined also by multiple papers/articles already published.
However, the theme is extremely broad and the author is put in difficulty as in a constrained space the author has to comprise several information, modeling and conclusions.
The approach is framed in what it is known as ”algorithmic information theory”, concretely referring to ”the naturally occurring computational system(NOCS)”, concept and model closer to the realities of complex adaptive systems, having ”the collective intelligence system” as prototype.
From the complex aspects on NOCS that have been or will be researched -interactions, semantics, formalization, stimuli etc. – the author presents ”the role of transients in NOCS” and insists on a particular phenomenon ”transient induced global response synchronization (TIGoRS)”.
Synchronization emerges as a global feature for complex adaptive systems, presenting, each time, variations and specific aspects.
For better understanding and greater accessibility of the contents proposed, I consider that the author should complete/modify the paper with the following:
- explicit presentation of the purpose/finality of the paper in the abstract;
-restraining the first part concerning the four approaches of information theory and greater focus on NOCS concept and model, with more examples:
-extending the subchapter ”Discussion”, in view to include also considerations on the variation of entropy related to different transients as well as suggestions concerning future researches on this topic.
Author Response
Thank you for your review.
In response to your concerns
1) I have modified the abstract to, I hope, make the focus of the paper clearer
2) I have added a section at the end of the introduction which expands the discussion of NOCS. Hopefully this provides a fuller understanding of the concept and the motivations behind its introduction. For more details the interested readers can explore the more detailed papers on this subject as listed in the references (these are my World Futures papers)
3) I have expanded the discussion section to make the implications of the use of transients clearer. I have commented on future directions. I did not study specifically the entropy of these transients. Although not specifically referenced in this paper, the other papers on TIGoRS referred to in the references do mention that a variety of different stimulus patterns were sampled during these trials. Different complex systems models appear to respond to different symmetry types. In some cases, patterns with very high levels of symmetry induce TIGoRS. These are low entropy patterns. In other cases, complex, chaotic or even pseudorandom patterns induced TIGoRS, These are all high entropy patterns. It is not clear whether there is any relationship between the entropy of a pattern and its ability to induce TIGoRs, or the entropy of pattern generated by an autonomous complex system and that of the patterns that induce TIGoRS in it. This would seem an interesting area for study. A paragraph to that effect was also added to the discussion section.
I hope that this addresses the concerns raised.